# Assessing stress, anxiety, and depression in children and adolescents: Validation of the DASS-Y in Poland

Andrzej Śliwerski ☯*, Karolina Koszałkowska ☯, Izabela Socha ☯

Institute of Psychology, University of Łódź, Łódź, Poland

☯ These authors contributed equally to this work.
* andrzej.sliwerski@uni.lodz.pl

## Abstract

### Background

The Depression Anxiety Stress Scales (DASS-21) is a widely used measure of negative emotional states in adults and adolescents. A youth version (DASS-Y) was recently developed and validated to address the developmental needs of younger populations. While the DASS-Y has shown promising psychometric properties in several languages, no Polish adaptation has been conducted to date. This study aimed to validate the Polish version of the DASS-Y and assess its psychometric performance in a sample of children and adolescents.

### Methods

A total of 344 children and adolescents aged 9–17 years were recruited using a school-based convenience sampling strategy between January 6 and February 16, 2025. Participants completed a sociodemographic questionnaire, the Polish version of the DASS-Y, the Children's Depression Inventory (CDI), and the Self-Assessment Manikin (SAM) scale.

### Results

CFA supported the original three-factor structure of the DASS-Y with satisfactory model fit indices. The subscales demonstrated excellent internal consistency, with Cronbach's alpha and McDonald's omega values exceeding 0.85. The DASS-Y subscales correlated significantly with the CDI and SAM scales in expected directions, supporting convergent and discriminant validity. Metric invariance across genders was established, though scalar invariance was not fully supported, indicating potential differences in item intercepts between boys and girls.

**Citation:** Śliwerski A, Koszałkowska K, Socha I (2025) Assessing stress, anxiety, and depression in children and adolescents: Validation of the DASS-Y in Poland. PLoS One 20(8): e0323835. https://doi.org/10.1371/journal.pone.0323835

**Data availability statement:** The data underlying the results presented in this study are publicly available in the OSF repository. URL: https://osf.io/zp5tk/ DOI: 10.17605/OSF.IO/ZP5TK

**Funding:** The author(s) received no specific funding for this work.

**Competing interests:** The authors have declared that no competing interests exist.

## Conclusions

The Polish version of the DASS-Y is a reliable and valid tool for assessing depression, anxiety, and stress in Polish children and adolescents. While the instrument functions similarly across genders in terms of structure, caution is advised when comparing mean scores between boys and girls due to partial scalar invariance. Future research should explore the DASS-Y's applicability in clinical settings and its sensitivity to changes over time.

## Introduction

Increasing rates of mood-related disorders across various populations call for the need to develop reliable, concise, and openly available psychometric tools for a quick assessment of symptoms related to depression, anxiety, and stress. One widely used tool for such measurement is the Depression Anxiety and Stress Scale (DASS) [1], a 42-item self-report instrument of a three-dimensional structure consisting of subscales of negative emotional states of depression, anxiety, and tension, each measured by 14 items. A parallelly available version of DASS-21 includes seven items per subscale, and both versions of DASS are appreciated for their psychometric soundness and ease of use for scientific, non-clinical, and clinical assessment [e.g., 2–4] across multiple countries and populations [e.g., 5,6]. Notably, the structure of DASS is partially in line with the Tripartite model of anxiety and depression [7], which posits that even though anxiety and depression share a common core component of psychological affective distress, each has specific factors that allow for the differentiation of specific anxiety (marked with hyperarousal) and specific depression (marked with anhedonia). Differently to the tripartite model, the creators of DASS observed a stable longitudinal psychometric pattern of a third specific set of symptoms related to tension and irritability [8], consisting of agitation, irritability, impatience, difficulty relaxing, and nervous arousal [1]. These three factors of depression, anxiety, and stress, although moderately intercorrelated, are not necessarily believed to overlap but rather share common biological and environmental causes. Moreover, the factorial structure of the DASS-21 has been supported in numerous studies across cultures and demographic groups, with evidence of measurement invariance allowing for meaningful comparisons between different populations [5,6].

In recent years, particularly following the COVID-19 pandemic and other global crises, there has been a growing concern regarding the mental health of more vulnerable populations, such as children and adolescents. For example, a recent report [9] by the Centers for Disease Control and Prevention in the USA found that in 2023, 40% of high school students felt sad or hopeless almost daily for at least two consecutive weeks, to the point of withdrawing from their usual activities, and 20% of them were seriously considering suicide. Trends for both phenomena increased between 2013 and 2023 and peaked around the pandemic period around 2021 [9]. Even before the pandemic, data suggests that in 2019, around 15% of European boys and girls aged 10–14 reported struggling with a mental disorder. For youth aged 15–19,

the prevalence was over 18% for girls and 16% for boys. Among these, anxiety and depression accounted for 55% of mental disorders among European adolescents aged 10–19 [10].

Since children and adolescents are especially vulnerable to mood disorders, and adult recurrences of child and adolescent symptoms related to negative affect are common [11,12], one would expect the three-factor structure of DASS to be replicated in younger populations. Moreover, most of the existing assessment tools targeted at youth are limited in their capacity to adequately measure specific symptom clusters that would indicate more anxiety-, stress- or depression-based disorders and instead rely on very general scores of negative affect or provide no items relating to co-occurring depression [13]. Szabó and Lovibond [14] attempted to test the existing adult DASS items with a sample of children aged 7–14, failing to differentiate between anxiety, stress, and depression, part of which was attributed to the item contents that might not have been specific enough to symptoms displayed by children. Further attempts, although of greater success in replicating the three-factor structure of DASS in adolescents between ages 12–18, were still not suitable for younger children [15]. To verify the generalizability of the three-factor structure of adult DASS in a more age-appropriate language and following the need to address the gap in tools available for children and adolescents that would be useful in discriminating between depression, anxiety, and tension [1] a 21-item Depression, Anxiety, and Stress Scale for Youth (DASS-Y) was developed by Szabó and Lovibond [14]. The instrument was designed to be presented to participants between 7 and 18 years of age and withheld its three-dimensional structure, supporting Clark and Watson's [7] tripartite model of anxiety and depression, as well as Lovibond's [1,8] notion about stress/tension as a separate group of distress indicators. Other languages that have already officially translated or culturally adapted DASS-Y include Chinese [16], Serbian [17], German, Hebrew, Vietnamese, Persian [18], Indonesian [19], Spanish, Dutch, and Hindi. Although most of these projects relied on adolescent rather than child samples, they managed to replicate the psychometric properties and hypothesized structure of the DASS-Y, and in the case of the Chinese adaptation, the DASS-Y showed even better discriminant validity than the adult version of the instrument [16]. In the case of the DASS-Y, initial studies examining measurement invariance have shown promising results. For example, recent cross-cultural investigations supported its invariance across gender and countries (such as Serbia and Australia), as well as among Chinese and Indonesian adolescents [16,17,19]. These findings indicate that the DASS-Y can be considered structurally stable and comparable across diverse adolescent populations. Given this growing evidence, we aimed to investigate whether the Polish version of the DASS-Y would demonstrate comparable reliability and validity in assessing depressive, anxiety, and stress-related characteristics of children and adolescents in a non-clinical convenience sample.

## Materials and methods

### Translation

The original DASS-Y items were translated from English to Polish and then back-translated from Polish to English by two psychologists with high English proficiency and expertise in psychometrics and clinical psychology. All translation differences were then resolved through discussion until equivalence was reached [20]. The entire content of the Polish version of DASS-Y is openly available on the DASS website https://www2.psy.unsw.edu.au/dass/DASS-Y%20Translations.htm.

### Data collection

The sample size for this study was determined based on recommendations suggesting a minimum participant-to-item ratio of 5:1 for confirmatory factor analysis (CFA) [21]. Given that the DASS-Y consists of 21 items, a minimum of 105 participants was required. We recruited 344 participants to ensure adequate statistical power for CFA and additional analyses.

We recruited young participants from four educational institutions: two primary schools, one primary school combined with a secondary school, and one secondary school. In each of these institutions, formal consent was obtained from the principal, which was one of the essential prerequisites for conducting the study. Parental consent was also obtained, and

parents received information about the study via the online student information system and the children themselves. The recruitment and data collection period took place between 06/01/2025 and 16/02/2025. We presented the participants with a battery of self-report, "paper-pencil" type questionnaires, and the entire data collection process was carried out by the research team, supported by the presence of a psychologist or school counselor, and lasted for two weeks. This time-frame allowed for a comprehensive understanding of the institutions' functioning and ensured the precise execution of the research procedures. Notably, not all students could participate in the study—approximately 567 parents did not provide consent for their children to participate, which affected the final sample composition.

Additionally, per the inclusion criteria, children who could not read fluently or did not have a proficient command of Polish were excluded from taking part in the study, ensuring the fullest possible comprehension of the provided content. Before participating in the study, each child reviewed the informed consent form, guaranteeing their voluntary and informed participation. After becoming familiar with the consent form, children were given the opportunity to ask questions and decide whether they wished to participate. None of the children declined participation. Subsequently, participants completed a sociodemographic survey followed by a battery of questionnaires. Each participant received clear instructions on how to complete the questionnaires. Among the entire sample, only one child refused to participate, while the rest engaged willingly and actively in completing the questionnaires, demonstrating a high level of acceptance of the study among young participants.

## Instruments

### Ethical consideration

The study was conducted in accordance with ethical guidelines. Permission to create a translation and adaptation of the DASS-Y was obtained from the original author, Peter Lovibond, in November 2023, and the adaptation was made publicly available on the official DASS website [22]. Ethical approval was granted by the relevant ethics committee (Approval No. 2/KEBN-UŁ/V/2023-24) at the University of Łódź to conduct research involving children. The study required written informed consent from both the child's legal guardian and the child on the day of participation, with the option to withdraw at any stage. Data was collected directly from participants whose parents provided consent, and all responses remained anonymous.

### Statistical analysis

All statistical analyses were conducted using RStudio 2024.12.1. Data preparation, factor analysis, and reliability assessments were performed using a range of R packages, including *lavaan* for confirmatory factor analysis (CFA), *psych* for reliability and correlation analyses, *cocor* for comparing dependent correlations, *ggcorrplot* for visualization of correlation matrices, and *car* for regression diagnostics. Descriptive statistics and the formatting of tables were generated using JASP.

To evaluate the factor structure of the Polish adaptation of the DASS-Y, CFA was conducted using the Robust Maximum Likelihood estimator (MLR), which provides robust standard errors and is less sensitive to violations of multivariate normality. Model fit was assessed using commonly recommended fit indices: the Comparative Fit Index (CFI) and the Tucker-Lewis Index (TLI), with values above 0.90 considered acceptable and above 0.95 indicating excellent fit. Additionally, the Root Mean Square Error of Approximation (RMSEA) and its 90% confidence interval (CI) were used, where values below 0.08 indicate an acceptable fit, and values below 0.06 suggest a good fit. The Standardized Root Mean Square Residual (SRMR) was also examined, with values below 0.08 considered acceptable.

Measurement invariance across gender was tested through a series of nested CFA models, progressing from configural to metric and scalar invariance. The chi-square difference test was used to compare these models, with non-significant differences indicating measurement invariance. Additionally, changes in CFI and RMSEA were considered, following current best practices to evaluate invariance more robustly.

McDonald's omega (ω) and Cronbach's alpha (α) were computed for each subscale and the total scale for internal consistency. Convergent and discriminant validity were assessed through Pearson correlations between the DASS-Y subscales, the Children's Depression Inventory (CDI), and the Manikin scales.

Predictive validity was evaluated using multiple regression analyses, with the CDI Total Score and Manikin subscales as outcome variables. To test discriminant validity, the strength of overlapping dependent correlations was compared using the Pearson-Filon z-test and the Williams t-test, implemented in the *cocor* package.

All significance tests were conducted at α = .05, and confidence intervals for regression coefficients and model fit indices were reported where applicable.

## Results

### Characteristics of the participants

The demographic characteristics of the participants are presented in Table 1. The sample consisted of 344 participants, with a mean age of 13.84 years (SD = 2.36, range = 9–17 years). Regarding gender distribution, 55.2% (n = 190) of the participants identified as female, 38.7% (n = 133) as male, and 5.5% (n = 19) did not disclose their gender. Data on gender were missing for two participants (0.6%). In terms of place of residence, the majority of participants (78.2%) reported living in a large city, while 14.5% resided in a small town and 6.7% in a rural area. Eighteen participants (5.2%) declared having a chronic illness. The sample was predominantly Polish (78.8%), followed by Ukrainian (18.9%). Additionally, there were two participants each (0.6%) from Belarus and England, and one participant each (0.3%) from Kazakhstan and Italy. All participants were fluent in Polish.

### Confirmatory factor analysis

In order to determine whether the dataset was suitable for factor analysis, the Kaiser-Meyer-Olkin (KMO) measure of sampling adequacy and Bartlett's test of sphericity were performed. The KMO index was 0.94, indicating excellent factorability

**Table 1. Demographic characteristics of the sample.** *Descriptive statistics for gender, age distribution, and place of residence among the study participants (N = 344).*

|  |  | Frequency | Percent | Valid Percent | Cumulative Percent |
|---|---|---|---|---|---|
| Gender | Boy | 190 | 55.23 | 55.56 | 55.56 |
|  | Girl | 133 | 38.66 | 38.89 | 94.44 |
|  | Not disclosed | 19 | 5.52 | 5.56 | 100 |
| Age | 9 | 17 | 4.94 | 4.97 | 4.97 |
|  | 10 | 10 | 2.91 | 2.92 | 7.90 |
|  | 11 | 37 | 10.76 | 10.82 | 18.71 |
|  | 12 | 45 | 13.08 | 13.16 | 31.87 |
|  | 13 | 49 | 14.24 | 14.33 | 46.20 |
|  | 14 | 29 | 8.43 | 8.48 | 54.68 |
|  | 15 | 40 | 11.63 | 11.70 | 66.37 |
|  | 16 | 64 | 18.61 | 18.71 | 85.09 |
|  | 17 | 51 | 14.83 | 14.91 | 100 |
| Place of residence | Large city | 269 | 78.20 | 78.66 | 78.66 |
|  | Small town | 50 | 14.54 | 14.62 | 93.28 |
|  | Rural area | 23 | 6.69 | 6.72 | 100 |
| Missing |  | 2 | 0.58 |  |  |
| Total |  | 344 | 100 |  |  |

of the data. Individual measures of sampling adequacy for each item ranged between 0.91 and 0.96, confirming that all items were appropriate for inclusion in factor analysis.

Furthermore, Bartlett's test of sphericity was statistically significant, $\chi^2$ (210) = 3812.01, p<0.001, indicating that the correlation matrix significantly differed from an identity matrix. This result suggests that the variables were sufficiently correlated to justify factor analysis.

Confirmatory Factor Analysis (CFA) was conducted to examine the factor structure of the Polish adaptation of the DASS-Y. The three-factor model, reflecting Stress, Anxiety, and Depression, was tested using the Robust Maximum Likelihood estimator (MLR). The model demonstrated acceptable fit, with CFI = 0.926 and TLI = 0.917, indicating a good fit to the data. The RMSEA was 0.057 (90% CI [0.050, 0.064]), and the SRMR was 0.056, both suggesting an adequate fit.

All items loaded significantly onto their respective factors, with standardized loadings ranging from 0.547 to 0.853 (see Figure 1). Since all factor loadings exceeded the recommended threshold of 0.40, the items demonstrated adequate contribution to their respective subscales, supporting the validity of the three-factor model. The latent factors were highly correlated, with inter-factor correlations of 0.756 between Stress and Anxiety, 0.689 between Stress and Depression, and 0.775 between Anxiety and Depression. These values suggest substantial relationships among the constructs while remaining below the recommended threshold of 0.85, indicating that the factors are related but not redundant. Full details on factor loadings are presented in Table 2.

To assess the validity of the three-factor model, standardized factor loadings were examined to determine the degree to which each item was associated with its respective latent factor and whether any cross-loadings indicated stronger associations with alternative factors (Table 2). The results showed that all items loaded most strongly onto their designated factors—stress, anxiety, or depression—while their loadings on the other two factors were lower. This pattern supports the distinctiveness of the three subscales.

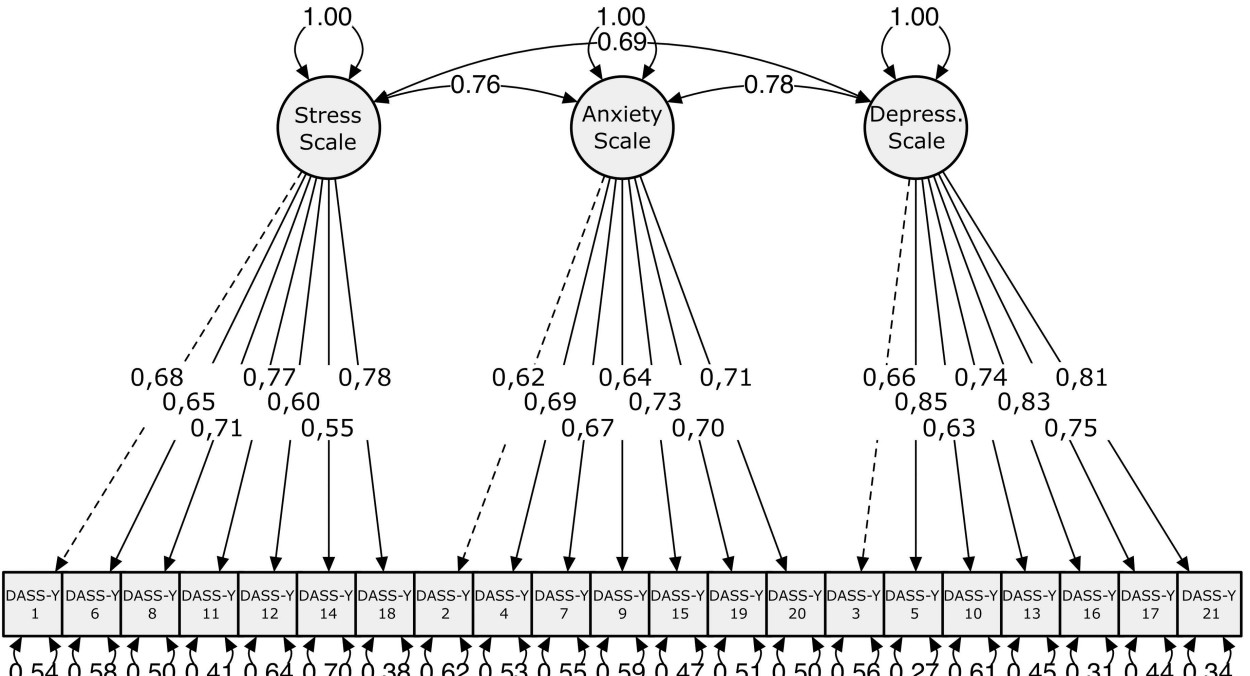

**Fig 1. Standardized Factor Loadings for the Three-Factor Model of the DASS-Y.**

**Table 2. Factor Loadings for the DASS-Y.** *Standardized Factor Loadings for the Three-Factor Model of the DASS-Y.*

| Factor | Indicator | Estimate | Std. Error | z-value | Std. loading | | |
|---|---|---|---|---|---|---|---|
| | | | | | Stress | Anxiety | Depression |
| DASS-Y | DASSY_1 | 1.000 | 0.000 | | 0.682 | | |
| Stress | DASSY_6 | 0.927 | 0.078 | 11.911*** | 0.651 | | |
| | DASSY_8 | 1.048 | 0.077 | 13.586*** | 0.710 | | |
| | DASSY_11 | 1.179 | 0.082 | 14.364*** | 0.767 | | |
| | DASSY_12 | 0.879 | 0.092 | 9.582*** | 0.597 | | |
| | DASSY_14 | 0.805 | 0.081 | 9.989*** | 0.547 | | |
| | DASSY_18 | 1.201 | 0.086 | 14.004*** | 0.788 | | |
| DASS-Y | DASSY_2 | 1.000 | 0.000 | | | 0.620 | |
| Anxiety | DASSY_4 | 1.001 | 0.082 | 12.171*** | | 0.687 | |
| | DASSY_7 | 1.068 | 0.097 | 10.956*** | | 0.672 | |
| | DASSY_9 | 0.980 | 0.099 | 9.941*** | | 0.637 | |
| | DASSY_15 | 1.059 | 0.106 | 9.978*** | | 0.728 | |
| | DASSY_19 | 1.143 | 0.103 | 11.075*** | | 0.703 | |
| | DASSY_20 | 0.979 | 0.094 | 10.425*** | | 0.706 | |
| DASS-Y | DASSY_3 | 1.000 | 0.000 | | | | 0.663 |
| Depression | DASSY_5 | 1.393 | 0.103 | 10.672*** | | | 0.853 |
| | DASSY_10 | 0.863 | 0.089 | 9.663*** | | | 0.625 |
| | DASSY_13 | 1.175 | 0.091 | 12.855*** | | | 0.743 |
| | DASSY_16 | 1.398 | 0.137 | 10.206*** | | | 0.832 |
| | DASSY_17 | 1.346 | 0.134 | 10.041*** | | | 0.745 |
| | DASSY_21 | 1.370 | 0.117 | 11.684*** | | | 0.814 |

**Std.Err** – Standard Error: Measures the variability of the estimate; **z-value** – z-score: The test statistic representing how far the estimate is from zero in standard error units; **Std.loading** – Standardized Loading in the Observed Variable Scale; *** < .001.

Regarding the stress factor, all items demonstrated strong primary loadings ranging from 0.547 (item 14) to 0.788 (item 18). Their cross-loadings on the anxiety and depression factors were substantially lower, confirming their specificity to the stress construct. Similarly, for the anxiety factor, the primary loadings varied between 0.620 (item 2) and 0.728 (item 15), while the cross-loadings remained consistently weaker, indicating a clear differentiation from the other subscales. Finally, the depression factor exhibited the strongest loadings among all three dimensions, with values ranging from 0.625 (item 10) to 0.853 (item 5), reinforcing the coherence of the depression subscale.

Overall, the factor loadings support the expected structure of the DASS-Y, with items strongly associated with their respective factors and minimal cross-loadings, confirming that the three-factor model provides a robust measurement of stress, anxiety, and depression in adolescents.

### Invariance testing across gender

The measurement invariance of the DASS-Y across genders was assessed using a series of nested confirmatory factor analysis (CFA) models. First, a configural invariance model was tested, allowing all parameters to vary freely between groups. This model demonstrated acceptable fit, indicating that the three-factor structure was consistent across gender groups. Next, metric invariance was tested by constraining factor loadings to be equal across groups. The chi-square difference test between the configural and metric models was not significant, $\Delta\chi^2$ (18) = 15.363, p = 0.637, suggesting that factor loadings were equivalent across gender. Changes in CFI and RMSEA between models did not exceed

recommended thresholds (ΔCFI < 0.01, ΔRMSEA < 0.015), supporting metric invariance. This result supports the assumption that the DASS-Y items relate to the latent constructs similarly for boys and girls.

Subsequently, scalar invariance was tested by additionally constraining item intercepts to be equal across groups. The chi-square difference test between the metric and scalar models was significant, $\Delta\chi^2$ (18) = 43.743, p = 0.0006, indicating that item intercepts differed between gender groups. This suggests that mean differences in observed scores cannot be solely attributed to differences in the latent constructs, as systematic variations in item responses may exist between boys and girls.

Further examination of modification indices revealed that item 12 in the stress scale exhibited the highest modification index (MI = 20.868), suggesting potential differences in factor loadings for this item between gender groups. A partial metric invariance model was tested to address this inconsistency, allowing this specific factor loading to vary across groups, which led to an improved model fit. Overall, the standardized factor loadings remained broadly comparable across boys and girls, as shown in Table 3, supporting metric invariance and indicating that items generally contribute similarly to the latent factors. However, despite these similarities in factor loadings, full scalar invariance was not supported. This result implies that observed mean differences in stress, anxiety, and depression scores may reflect not only actual variations in the latent constructs but also differences in how each group interpreted certain items. Therefore, future research should explore whether differential item functioning (DIF) affects the comparability of scores and consider whether separate norms should be established for different gender groups.

To further examine potential item-level differences, we compared the standardized factor loadings for boys and girls. As shown in Table 3, the loadings were broadly comparable across gender groups, with no substantial discrepancies. This

**Table 3. Standardized Factor Loadings for Boys and Girls.** *Standardized factor loadings for each DASS-Y item are presented separately for boys and girls.*

| Factor | Indicator | Boys (Std.loading) | Girls (Std.loading) |
|---|---|---|---|
| DASSY | DASSY_1 | 0.65 | 0.59 |
| Stress | DASSY_6 | 0.64 | 0.61 |
| | DASSY_8 | 0.69 | 0.63 |
| | DASSY_11 | 0.74 | 0.72 |
| | DASSY_12 | 0.60 | 0.54 |
| | DASSY_14 | 0.62 | 0.53 |
| | DASSY_18 | 0.80 | 0.74 |
| DASSY | DASSY_2 | 0.53 | 0.61 |
| Anxiety | DASSY_4 | 0.66 | 0.69 |
| | DASSY_7 | 0.65 | 0.73 |
| | DASSY_9 | 0.63 | 0.57 |
| | DASSY_15 | 0.69 | 0.72 |
| | DASSY_19 | 0.69 | 0.70 |
| | DASSY_20 | 0.66 | 0.75 |
| DASSY | DASSY_3 | 0.70 | 0.56 |
| Depression | DASSY_5 | 0.84 | 0.83 |
| | DASSY_10 | 0.65 | 0.62 |
| | DASSY_13 | 0.77 | 0.71 |
| | DASSY_16 | 0.82 | 0.79 |
| | DASSY_17 | 0.73 | 0.70 |
| | DASSY_21 | 0.80 | 0.80 |

**Std. loading** – Standardized Loading in the Observed Variable Scale; ***<.001

visual inspection supports the metric invariance findings, indicating that each item contributes similarly to the underlying factors in both groups. However, some minor variations should be interpreted cautiously and may guide future refinements or targeted analyses.

### Reliability

The internal consistency of the DASS-Y was assessed using McDonald's omega ($\omega$) and Cronbach's alpha ($\alpha$) coefficients for each subscale and the total score. As presented in Table 4, all subscales demonstrated high reliability, with omega values ranging from 0.856 (Anxiety) to 0.907 (Depression) and alpha values ranging from 0.854 (Stress) to 0.902 (Depression). The total scale exhibited excellent internal consistency ($\omega = 0.930$, $\alpha = 0.936$), indicating strong reliability across all items. These results suggest that the DASS-Y provides stable and consistent measurements of stress, anxiety, and depression in the study sample.

### Convergent and discriminant validity

A correlation analysis was conducted to assess the convergent and discriminant validity of the DASS-Y. Convergent validity was examined by analyzing the correlations between the DASS-Y subscales and the CDI (depression indicators) as well as the Manikin scales (assessing physiological arousal, emotional intensity, and perceived distress).

The DASS-Y stress, anxiety, and depression subscales demonstrated strong correlations with the CDI total score ($r = 0.618$–$0.809$), confirming convergent validity in measuring general negative affect. The strongest correlation was observed between the DASS-Y depression subscale and the CDI total score ($r = 0.809$), indicating that both scales measure closely related aspects of depressive symptomatology.

Analyzing specific CDI subscales, DASS-Y depression exhibited the highest correlations with CDI negative self-perception ($r = 0.787$) and CDI negative mood ($r = 0.698$), indicating strong alignment between depressive constructs in both instruments. In contrast, DASS-Y stress and anxiety showed comparatively lower correlations with CDI subscales, suggesting greater distinction from depressive aspects.

The DASS-Y subscales were compared with the Manikin scales: Physical Arousal, Emotional Intensity, and Perceived Distress to assess discriminant validity. While all correlations were statistically significant, they were substantially lower than those observed with the CDI. Physical Arousal was moderately correlated with Depression ($r = 0.623$) and Anxiety ($r = 0.534$ and to a lesser extent with Stress ($r = 0.433$), which is consistent with the physiological underpinnings of anxious and depressive states. Emotional Intensity was associated most strongly with Stress ($r = 0.342$), followed by Anxiety ($r = 0.333$), and only weakly with Depression ($r = 0.242$). Perceived Distress showed the weakest correlations with all three subscales, with the strongest being for Stress ($r = 0.268$) (see Fig 2).

To further evaluate discriminant validity, a test of overlapping dependent correlations was conducted comparing the strength of the association between DASS-Y Depression and CDI Total ($r = 0.81$) with the association between DASS-Y Depression and Manikin Physical Arousal ($r = 0.66$). The difference in correlation was statistically significant, $\Delta r = 0.15$, 95% CI [0.10, 0.21], $z = 5.54$, $p < .001$. This supports the interpretation that the DASS-Y Depression scale captures a construct more closely aligned with depression than with physiological arousal.

**Table 4. Internal Consistency of the DASS-Y.** *Reliability coefficients (McDonald's omega and Cronbach's alpha) for the DASS-Y subscales and total score.*

|  | Coefficient ω | Coefficient α |
|---|---|---|
| DASS-Y Stress | 0.865 | 0.854 |
| DASS-Y Anxiety | 0.856 | 0.855 |
| DASS-Y Depression | 0.907 | 0.902 |
| Total | 0.930 | 0.936 |

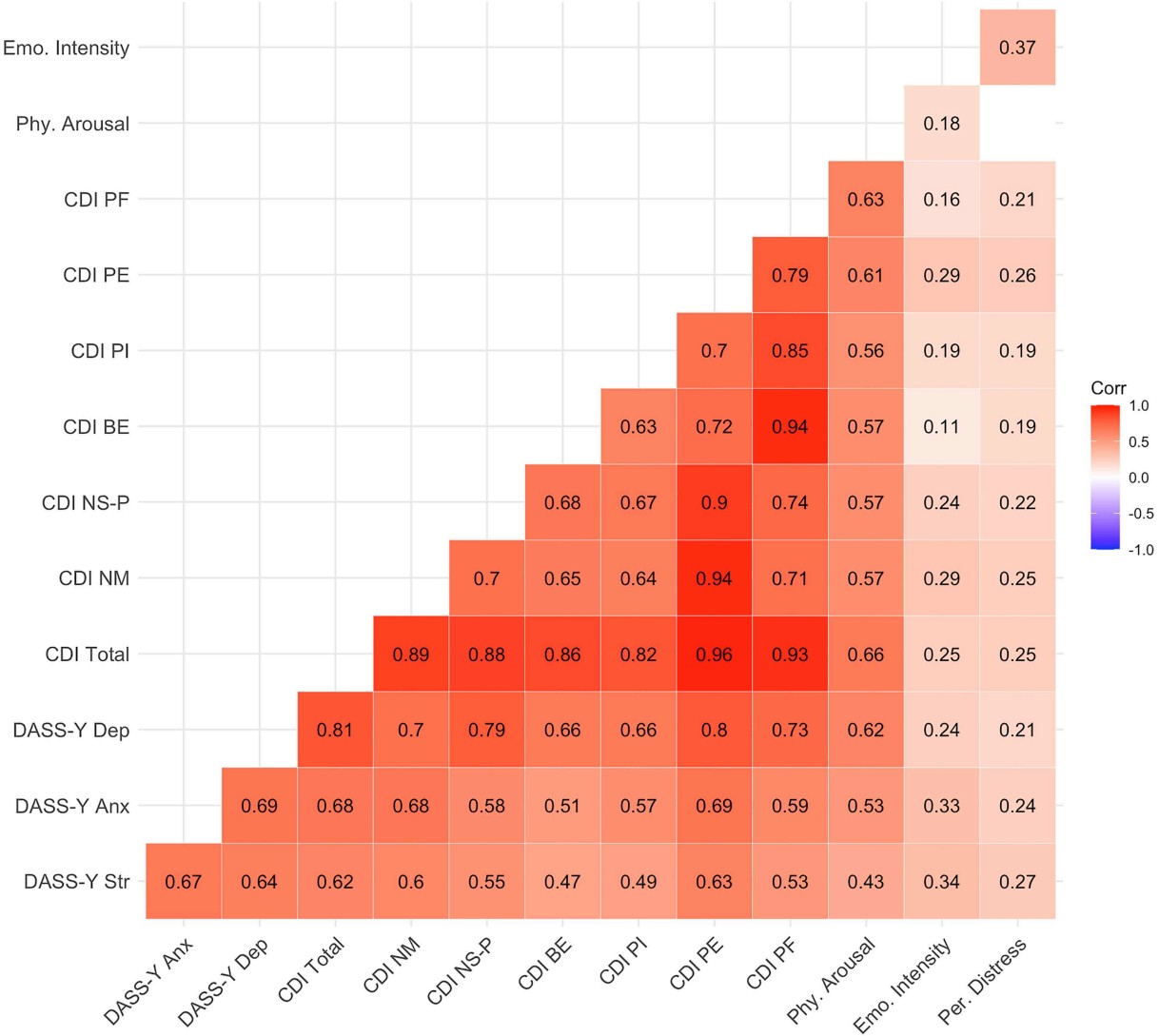

**Fig 2. Correlation Matrix of DASS-Y, CDI, and Manikin Scales.** Heatmap displaying the correlations between the DASS-Y subscales, the CDI subscales, and the Manikin dimensions. Darker shades indicate stronger correlations, with numerical values representing Pearson correlation coefficients. All displayed correlations are statistically significant (p < .05). **DASS-Y Str** – DASS-Y Stress; **DASS-Y Anx** – DASS-Y Anxiety; **DASS-Y Dep** – DASS-Y Depression; **CDI Total** – CDI Total score; **CDI NM** – CDI Negative Mood; **CDI NS-P** – CDI Negative Self-Perceptions; **CDI BE** – CDI Behavioral Engagement; **CDI PI** – CDI Personal Ineffectiveness; **CDI PE** – CDI Physiological Experiences; **CDI PF** – CDI Peer and Family Relations; **Phy. Arousal** – Manikin Physical Arousal dimension; **Emo. Intensity** – Manikin Emotional Intensity dimension; **Per. Distress** – Manikin Perceived Distress dimension.

Together, these results provide strong evidence for the convergent and discriminant validity of the DASS-Y in the present sample. The Depression subscale, in particular, shows high alignment with established depressive indicators and lower associations with distinct affective dimensions such as physiological arousal and perceived distress.

## Predictive validity

The predictive validity of the DASS-Y subscales was determined by multiple regression analyses conducted using the CDI Total score and the three Manikin scale dimensions as outcome variables (see Table 5). Before presenting regression

results, we assessed the assumptions of linear regression. The residuals plot did not reveal any strong patterns, indicating acceptable linearity. The Q-Q plot suggested a slight right skewness; however, this deviation was minor and common in large samples. The Shapiro-Wilk test indicated a deviation from normality ($W = 0.98$, $p < 0.001$), but given the large sample size, this was not deemed to impact the results substantially. The Breusch-Pagan test revealed heteroscedasticity ($\chi^2 = 6.72$, $p = 0.010$), suggesting that standard errors should be interpreted with caution.

The results indicate that the CDI Total Score, which assesses depressive symptoms, was significantly predicted by all three DASS-Y subscales, with Depression scale being the strongest predictor ($\beta = 0.61$, $p < .001$), followed by Anxiety scale ($\beta = 0.19$, $p < .001$) and Stress scale ($\beta = 0.10$, $p = .025$). The overall model explained 68.8% of the variance in CDI

**Table 5. Multiple Regression Analyses Predicting CDI Total Score and Manikin Scale Dimensions from DASS-Y Subscales.**

| Predictor | b | b 95% CI [LL, UL] | beta | beta 95% CI [LL, UL] | sr2 | sr2 95% CI [LL, UL] | r | Fit |
|---|---|---|---|---|---|---|---|---|
| **CDI Total Score as the criterion** | | | | | | | | |
| (Intercept) | 33.28** | [32.04, 34.52] | | | | | | |
| DASS-Y Str | 0.09* | [0.01, 0.17] | 0.10 | [0.01, 0.18] | .00 | [-.00,.01] | .62** | |
| DASS-Y Anx | 0.19** | [0.10, 0.29] | 0.19 | [0.10, 0.28] | .02 | [.00,.03] | .68** | |
| DASS-Y Dep | 0.57** | [0.49, 0.65] | 0.61 | [0.53, 0.70] | .18 | [.12,.23] | .81** | |
| | | | | | | | | $R^2 = .688^{**}$ |
| | | | | | | | | 95% CI[.64,.73] |
| **Manikin Physical Arousal dimension as the criterion** | | | | | | | | |
| (Intercept) | 1.50** | [1.34, 1.67] | | | | | | |
| DASS-Y Str | −0.00 | [-0.01, 0.01] | −0.02 | [-0.14, 0.10] | .00 | [-.00,.00] | .43** | |
| DASS-Y Anx | 0.02** | [0.01, 0.03] | 0.20 | [0.08, 0.33] | .02 | [-.00,.04] | .53** | |
| DASS-Y Dep | 0.04** | [0.03, 0.06] | 0.49 | [0.37, 0.61] | .11 | [.06,.17] | .62** | |
| | | | | | | | | $R^2 = .408^{**}$ |
| | | | | | | | | 95% CI[.33,.47] |
| **Manikin Emotional Intensity dimension as the criterion** | | | | | | | | |
| (Intercept) | 2.35** | [2.11, 2.59] | | | | | | |
| DASS-Y Str | 0.03** | [0.01, 0.04] | 0.23 | [0.09, 0.38] | .03 | [-.00,.06] | .34** | |
| DASS-Y Anx | 0.03** | [0.01, 0.04] | 0.21 | [0.06, 0.37] | .02 | [-.01,.05] | .33** | |
| DASS-Y Dep | −0.01 | [-0.02, 0.01] | −0.06 | [-0.20, 0.09] | .00 | [-.01,.01] | .24** | |
| | | | | | | | | $R^2 = .138^{**}$ |
| | | | | | | | | 95% CI[.07,.20] |
| **Manikin Perceived Distress dimension as the criterion** | | | | | | | | |
| (Intercept) | 2.48** | [2.24, 2.72] | | | | | | |
| DASS-Y Str | 0.02* | [0.00, 0.03] | 0.18 | [0.04, 0.33] | .02 | [-.01,.04] | .27** | |
| DASS-Y Anx | 0.01 | [-0.01, 0.03] | 0.11 | [-0.05, 0.27] | .01 | [-.01,.02] | .24** | |
| DASS-Y Dep | 0.00 | [-0.01, 0.02] | 0.02 | [-0.13, 0.17] | .00 | [-.00,.00] | .21** | |
| | | | | | | | | $R^2 = .079^{**}$ |
| | | | | | | | | 95% CI[.03,.13] |

*Note.* A significant *b*-weight indicates the beta-weight and semi-partial correlation are also significant. *b* represents unstandardized regression weights. *beta* indicates the standardized regression weights. $sr^2$ represents the semi-partial correlation squared. *r* represents the zero-order correlation. *LL* and *UL* indicate the lower and upper limits of a confidence interval, respectively. * indicates $p < .05$. ** indicates $p < .01$. **DASS-Y Str** – DASS-Y Stress; **DASS-Y Anx** – DASS-Y Anxiety; **DASS-Y Dep** – DASS-Y Depression.

Total Score (R$^2$ = .688, p < .001), indicating strong predictive power of the DASS-Y, particularly its Depression scale, for depressive symptoms measured by the CDI.

For the Manikin Physical Arousal dimension, which captures physiological activation associated with stress and anxiety, the regression model explained 40.8% of the variance (R$^2$ = .408, p < .001). Depression scale was the strongest predictor (β = 0.49, p < .001), followed by Anxiety scale (β = 0.20, p = .001), while Stress scale did not significantly contribute to the prediction (β = −0.02, p = .762).

The Manikin Emotional Intensity dimension, which reflects subjective emotional reactivity, was predicted primarily by Anxiety scale (β = 0.21, p = .006) and Stress scale (β = 0.23, p = .001). In contrast, the Depression scale was not a significant predictor (β = −0.06, p = .457). However, the variance explained by the model was modest (R$^2$ = .138, p < .001).

Similarly, the Manikin Perceived Distress dimension, representing the subjective experience of distress, was significantly predicted only by Stress scale (β = 0.18, p = .015), while neither Anxiety scale (β = 0.11, p = .169) nor Depression scale (β = 0.02, p = .814) contributed significantly. The overall model explained 7.9% of the variance (R$^2$ = .079, p < .001).

## Discussion

The present study aimed to validate the Polish adaptation of the Depression Anxiety Stress Scales for Youth (DASS-Y) in a sample of children and adolescents aged 9–17 years. The results supported the original three-factor structure of the DASS-Y, reflecting distinct but related dimensions of stress, anxiety, and depression. Confirmatory factor analysis demonstrated satisfactory model fit, with all standardized factor loadings exceeding the recommended threshold of 0.40. The internal consistency of each subscale, as measured by both Cronbach's alpha and McDonald's omega, was high, further supporting the instrument's reliability.

These findings are consistent with previous validations of the DASS-Y. The original study by Szabó and Lovibond [14] reported a clear three-factor model, which was also confirmed in cross-cultural adaptations, such as the Indonesian version [19] and the Serbian version [17]. Similarly to those studies, the strongest factor loadings in the present sample were observed on the Depression subscale. The substantial inter-factor correlations remained below the .85 threshold, indicating that the scales measure related but distinct constructs. Moreover, the internal consistency of the Polish version was excellent, with both Cronbach's alpha and McDonald's omega exceeding .85 for all three subscales and surpassing .90 for the total score. These results align closely with prior studies, confirming that the DASS-Y maintains robust psychometric properties across diverse cultural contexts and provides a reliable assessment of depression, anxiety, and stress in children and adolescents.

The meaningful correlations between the DASS-Y and related psychological constructs supported the theoretical validity of the tool. As expected, the DASS-Y Depression subscale showed the strongest association with the CDI, a well-established measure of depressive symptoms. In contrast, the Anxiety and Stress subscales demonstrated distinct patterns of association with emotional dimensions captured by the Manikin scales. The Manikin Physical Arousal dimension was primarily predicted by DASS-Y Depression and Anxiety, while Emotional Intensity was associated with both DASS-Y Anxiety and Stress. On the other hand, Perceived Distress was uniquely related to the Stress subscale. Notably, neither Emotional Intensity nor Perceived Distress showed significant associations with the Depression subscale, suggesting that these dimensions reflect physiological and affective arousal rather than cognitive or motivational aspects of depression. These patterns support the distinctiveness of the DASS-Y subscales and their alignment with theoretically grounded models of emotional functioning in youth.

The present study also examined measurement invariance across genders to ensure that the DASS-Y assesses the same constructs in boys and girls. The results supported metric invariance, indicating that the factor loadings were equivalent across gender groups, allowing for meaningful comparisons of relationships between latent variables. However, scalar invariance was not fully supported, suggesting that observed score differences may not reflect actual differences in latent means, particularly for specific items such as item 12, which showed differential intercepts. These findings partially align with previous research. For instance, Jovanović [17] reported both metric and scalar invariance across gender in a

Serbian sample and scalar invariance across national groups (Serbia and Australia), highlighting the scale's potential for cross-cultural application. While the current study confirms that the DASS-Y functions similarly across genders in terms of structure, caution is advised when interpreting mean score differences, especially for the stress subscale.

While the findings of this study offer strong support for the validity and reliability of the Polish version of the DASS-Y, several considerations should be taken into account. First, the sample, although diverse in age and geographic background, was based on convenience sampling. Although this does not undermine the internal validity of the findings, future research could benefit from larger and more representative samples to enhance the generalizability of results across different regions and socio-economic groups.

Second, the study relied exclusively on self-report measures. While the DASS-Y was developed with developmental appropriateness in mind, younger children may face challenges in accurately articulating their emotional experiences. Including parent or teacher reports or interview-based assessments in future studies may complement self-report data and provide a more comprehensive picture.

Third, although the internal consistency of the subscales was excellent, test-retest reliability was not assessed. Future research could evaluate the temporal stability of the DASS-Y scores to confirm its suitability for monitoring changes over time or in longitudinal designs.

Finally, using the CDI as a criterion measure allowed for a solid assessment of convergent validity, but the study did not include clinical diagnostic interviews. Incorporating structured clinical assessments in future validation studies could provide a stronger basis for evaluating the tool's diagnostic sensitivity and specificity. In addition, minor violations of regression assumptions, including heteroscedasticity and slight deviations from normality of residuals, suggest that future studies should consider robust estimation techniques when conducting predictive analyses.

## Conclusions

The present study provides robust evidence for the validity and reliability of the Polish adaptation of the DASS-Y in a sample of children and adolescents aged 9–17. The three-factor structure of stress, anxiety, and depression was confirmed through confirmatory factor analysis, and the scale demonstrated high internal consistency. Evidence for convergent, discriminant, and predictive validity further supports the theoretical soundness of the instrument.

Notably, the DASS-Y proved to be a brief yet psychometrically sound tool for assessing negative emotional states in youth. Its suitability for use in both research and educational or clinical screening settings is particularly noteworthy, as it allows for the simultaneous assessment of three distinct but related domains of distress.

While some limitations remain, including the partial lack of scalar invariance and the absence of clinical diagnostic data, the Polish version of the DASS-Y can serve as a valuable instrument for identifying emotional difficulties in children and adolescents. Future research may extend its application in clinical populations and explore its usefulness in monitoring treatment outcomes or emotional well-being in school-based prevention programs.

## Acknowledgments

We want to express our sincere gratitude to Magdalena Koronowska-Miksa, Wiktor Florczyk, Agnieszka Szubska, and Kamila Bankowska for their invaluable support and assistance during the course of this study. We are also deeply grateful to the parents who provided consent to their children's participation and, most importantly, to the children and adolescents who took part in the study.

## Author contributions

**Conceptualization:** Andrzej Śliwerski, Karolina Koszałkowska.

**Data curation:** Izabela Socha.

**Formal analysis:** Andrzej Śliwerski.

**Investigation:** Izabela Socha.

**Methodology:** Andrzej Śliwerski, Karolina Koszałkowska.

**Project administration:** Andrzej Śliwerski.

**Resources:** Izabela Socha, Karolina Koszałkowska.

**Software:** Andrzej Śliwerski.

**Supervision:** Andrzej Śliwerski.

**Validation:** Karolina Koszałkowska.

**Visualization:** Andrzej Śliwerski.

**Writing – original draft:** Andrzej Śliwerski, Izabela Socha, Karolina Koszałkowska.

**Writing – review & editing:** Andrzej Śliwerski, Karolina Koszałkowska.

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
