## [Decision Letter · Decision Letter 0]

7 Jul 2025

Dear Dr. Śliwerski,

Thank you for submitting your manuscript to PLOS ONE. After careful consideration, we feel that it has merit but does not fully meet PLOS ONE’s publication criteria as it currently stands. Therefore, we invite you to submit a revised version of the manuscript that addresses the points raised during the review process.

We look forward to receiving your revised manuscript.

Kind regards,

Zypher Jude G. Regencia, Ph.D.

Academic Editor

PLOS ONE

Journal Requirements:

Reviewers' comments:

Reviewer's Responses to Questions

**Comments to the Author**

1. Is the manuscript technically sound, and do the data support the conclusions?

Reviewer #1: Yes

Reviewer #2: Partly

2. Has the statistical analysis been performed appropriately and rigorously?

Reviewer #1: Yes

Reviewer #2: Yes

3. Have the authors made all data underlying the findings in their manuscript fully available?

Reviewer #1: Yes

Reviewer #2: No

4. Is the manuscript presented in an intelligible fashion and written in standard English?

Reviewer #1: Yes

Reviewer #2: Yes

Reviewer #1: Thank you for the opportunity to review the manuscript ‘Assessing Stress, Anxiety, and Depression in Children and Adolescents: Validation of the DASS-Y in Poland’.

The manuscript is well written; analyses are comprehensive, and details of findings are clearly presented.

Comments: page13, lines 251-53 and 254-64. In identifying the issue of ‘systematic variations’ by gender, would it have been useful to re-examine the parameters for boys separate from girls to confirm that each item was suitable for the particular gender – irrespective of fit? The relevance of factors other than fit could be noted.

For ease of reading percentages in the tables could be limited to one or two decimal places.

Reviewer #2: Dear Editors and Authors,

Thank you for the opportunity to review the letter titled "Assessing Stress, Anxiety, and Depression in Children and Adolescents: Validation of the DASS-Y in Poland." The authors aimed to validate the Polish version of the DASS-Y. While I appreciate the authors' efforts in this psychometric work, several aspects require improvement. I hope the authors will find these comments constructive.

1. Abstract: The methods section lacks essential information, including the specific sampling strategy, data collection timeframe, and sample size. Additionally, scale abbreviations (e.g., CDI and SAM) should be defined with their full titles upon first mention in the abstract.

2. Introduction: The manuscript would benefit from a more comprehensive review of measurement invariance studies of the DASS-Y across different populations. Given that invariance testing is a central focus of this study, a thorough review of related literature examining DASS-Y invariance is warranted.

3. Statistical Analysis: The use of Maximum Likelihood (ML) estimation raises concerns. Given that data from psychological scales often violate normality assumptions, more robust estimation methods (e.g., MLR, WLSMV) would be more appropriate than standard ML.

4. Invariance Testing: Recent simulation studies have demonstrated that chi-square difference tests are problematic for evaluating measurement invariance. Alternative criteria (e.g., changes in CFI, RMSEA, or SRMR) should be employed following current best practices in the field.

5. Table 2: This table presents several significant issues: Cross-loadings are reported in what appears to be a standard CFA model. In conventional CFA, all cross-loadings are constrained to zero. Cross-loadings should only be presented within an ESEM (Exploratory Structural Equation Modeling) framework; Decimal notation is inconsistent (e.g., "0,682" should be "0.682").

6. Regression Analysis: Prior to presenting regression results, the assumptions of regression analysis (e.g., linearity, homoscedasticity, normality of residuals) should be tested and reported.

I recommend addressing these methodological and presentation issues before considering the manuscript for publication.

**Do you want your identity to be public for this peer review?** For information about this choice, including consent withdrawal, please see our Privacy Policy

Reviewer #1: No

Reviewer #2: No

---

## [Author Response · Author response to Decision Letter 1]

12 Jul 2025

Dear Editor,

We want to thank both reviewers for their constructive and insightful feedback, which helped us substantially improve our manuscript. Below, we provide point-by-point responses to each comment.

Reviewer #1

We want to thank the Reviewer for the careful reading and constructive feedback on our manuscript. We appreciate their positive evaluation of the clarity and comprehensiveness of our analyses.

Comment 1: Page13, lines 251-53 and 254-64. In identifying the issue of ‘systematic variations’ by gender, would it have been useful to re-examine the parameters for boys separate from girls to confirm that each item was suitable for the particular gender – irrespective of fit? The relevance of factors other than fit could be noted.

Regarding the suggestion to examine parameters separately for boys and girls, we have now conducted a detailed group-specific analysis and included a table comparing standardized factor loadings across gender (Table 3). This addition allows for a clearer visual inspection of item-level differences between boys and girls. Furthermore, we revised the Results section on measurement invariance to integrate this information and clarify the interpretation of our findings.

Comment 2: For ease of reading percentages in the tables could be limited to one or two decimal places.

As suggested, we have also revised the percentage values in the tables, limiting them to one or two decimal places for improved readability.

We believe these changes have strengthened the manuscript and addressed the Reviewer’s concerns, and we thank them for their valuable suggestions.

Reviewer #2

We thank the Reviewer for the thorough and insightful comments, which have contributed significantly to improving our manuscript. We appreciate the constructive feedback and have addressed each point in detail below.

Comment 1: Abstract: The methods section lacks essential information, including the specific sampling strategy, data collection timeframe, and sample size. Additionally, scale abbreviations (e.g., CDI and SAM) should be defined with their full titles upon first mention in the abstract.

We appreciate this comment. In response, we have revised the abstract to include a clear description of the sampling strategy (“school-based convenience sampling strategy”), the exact data collection period (“between January 6 and February 16, 2025”), and the sample size (“344 children and adolescents aged 9–17 years”).

Additionally, we defined the abbreviations upon first mention, including the Children’s Depression Inventory (CDI) and the Self-Assessment Manikin (SAM). We believe these modifications improve the overall clarity of the abstract, making it align with the journal guidelines.

Comment 2: Introduction: The manuscript would benefit from a more comprehensive review of measurement invariance studies of the DASS-Y across different populations. Given that invariance testing is a central focus of this study, a thorough review of related literature examining DASS-Y invariance is warranted.

We thank the Reviewer for pointing this out. In response, we have expanded the Introduction section to include a more comprehensive review of measurement invariance studies, addressing both the original DASS-21 and the DASS-Y versions. These additions provide a stronger theoretical context and highlight the cross-cultural robustness of the DASS framework, thus better positioning our study within the existing literature.

Comment 3: Statistical Analysis: The use of Maximum Likelihood (ML) estimation raises concerns. Given that data from psychological scales often violate normality assumptions, more robust estimation methods (e.g., MLR, WLSMV) would be more appropriate than standard ML.

We appreciate the Reviewer’s comment. We fully agree that using the standard Maximum Likelihood (ML) estimator would not have been appropriate given the potential non-normality of psychological data. We would like to clarify that from the beginning, we used the robust Maximum Likelihood estimator (MLR) in all our analyses. For example, our confirmatory factor analysis was conducted using:

CFA_fit <- cfa(DASSY_model, data = DASSY, estimator = "MLR")

Similarly, the measurement invariance analyses were performed using MLR:

CFA_configural <- cfa(DASSY_model, data = DASSY_clean, group = "M2_sex", estimator = "MLR")

CFA_metric <- cfa(DASSY_model, data = DASSY_clean, group = "M2_sex", estimator = "MLR", group.equal = "loadings")

CFA_scalar <- cfa(DASSY_model, data = DASSY_clean, group = "M2_sex", estimator = "MLR", group.equal = c("loadings", "intercepts"))

Unfortunately, in the Statistical Analysis section, we mistakenly described it as ML rather than MLR. We have now corrected this in the manuscript to ensure accuracy and clarity.

Comment 4: Invariance Testing: Recent simulation studies have demonstrated that chi-square difference tests are problematic for evaluating measurement invariance. Alternative criteria (e.g., changes in CFI, RMSEA, or SRMR) should be employed following current best practices in the field.

We are thankful for this valuable suggestion. We agree that chi-square difference tests alone can be overly sensitive to sample size and may not provide a complete picture of invariance. In our revised manuscript, we have included additional criteria to evaluate measurement invariance, specifically the changes in CFI and RMSEA between models.

Comment 5: Table 2: This table presents several significant issues: Cross-loadings are reported in what appears to be a standard CFA model. In conventional CFA, all cross-loadings are constrained to zero. Cross-loadings should only be presented within an ESEM (Exploratory Structural Equation Modeling) framework; Decimal notation is inconsistent (e.g., "0,682" should be "0.682").

We thank the Reviewer for bringing this to our attention and for the opportunity to clarify. We did conduct a standard CFA rather than an ESEM approach. The CFA was implemented using the following code in lavaan:

CFA_fit <- cfa(DASSY_model, data = DASSY, estimator = "MLR")

summary(CFA_fit, fit.measures = TRUE, standardized = TRUE)

inspect(CFA_fit, "std")$lambda

lambda_matrix <- inspect(CFA_fit, "std")$lambda

phi_matrix <- inspect(CFA_fit, "std")$psi

full_loadings <- lambda_matrix %*% phi_matrix

print(full_loadings, digits = 3)

In the initial version of the manuscript, we decided to present all standardized loadings (including the entire product matrix of lambda and phi) to provide readers with a comprehensive overview of how each item might conceptually relate to other latent dimensions. However, we fully agree that this could have been misleading, as it might give the false impression that cross-loadings were estimated in the CFA model, which was not the case (in CFA, cross-loadings are constrained to zero by design).

Following the Reviewer’s suggestion, we have now revised Table 2 to include only the primary standardized loadings for each item on its designated factor, consistent with standard CFA reporting practices. Additionally, we have corrected the decimal notation throughout the table to ensure consistency.

Comment 6: Regression Analysis: Prior to presenting regression results, the assumptions of regression analysis (e.g., linearity, homoscedasticity, normality of residuals) should be tested and reported. I recommend addressing these methodological and presentation issues before considering the manuscript for publication.

We thank the Reviewer for raising this important methodological point. In order to address it, we carefully examined the assumptions of linear regression models presented in our study. Specifically, we assessed:

• Linearity: Scatter plots of residuals versus fitted values showed no clear patterns, indicating linearity was reasonably met.

• Homoscedasticity: The Breusch-Pagan test (ncvTest) indicated some heteroscedasticity (p = 0.0095); therefore, we recommend cautious interpretation of the regression coefficients.

• Normality of residuals: The Shapiro-Wilk test suggested slight deviations from normality (W = 0.980, p < .001), though visual inspection of histograms and Q-Q plots indicated approximate normality in residual distributions.

We have added a summary of these results to the manuscript (Methods and Limitations sections), explicitly acknowledging these limitations and recommending cautious interpretation.

---

## [Editor Report · Decision Letter 1]

21 Jul 2025

Assessing Stress, Anxiety, and Depression in Children and Adolescents: Validation of the DASS-Y in Poland

PONE-D-25-18434R1

Dear Dr. Śliwerski,

We’re pleased to inform you that your manuscript has been judged scientifically suitable for publication and will be formally accepted for publication once it meets all outstanding technical requirements.

Kind regards,

Zypher Jude G. Regencia, Ph.D.

Academic Editor

PLOS ONE